# Current and Future Perspectives for Chimeric Antigen Receptor T Cells Development in Poland

**DOI:** 10.3390/biomedicines10112912

**Published:** 2022-11-13

**Authors:** Tomasz J. Ślebioda, Marcin Stanisławowski, Lucyna Kaszubowska, Jan M. Zaucha, Michał A. Żmijewski

**Affiliations:** 1Department of Histology, Medical University of Gdańsk, Dębinki 1, 80-210 Gdańsk, Poland; 2Department of Haematology and Transplantology, Medical University of Gdańsk, Smoluchowskiego 17, 80-214 Gdańsk, Poland

**Keywords:** CAR-T, CAR-NK, chimeric antigen receptor, tisagenlecleucel, axicabtagene ciloleucel, brexucabtagene autoleucel, ciltacabtagene autoleucel, idecabtagene vicleucel, lisocabtagene maraleucel, T cells

## Abstract

Chimeric antigen receptor T (CAR-T) cells are genetically modified autologous T cells that have revolutionized the treatment of relapsing and refractory haematological malignancies. In this review we present molecular pathways involved in the activation of CAR-T cells, describe in details the structures of receptors and the biological activity of CAR-T cells currently approved for clinical practice in the European Union, and explain the functional differences between them. Finally, we present the potential for the development of CAR-T cells in Poland, as well as indicate the possible directions of future research in this area, including novel modifications and applications of CAR-T cells and CAR-natural killer (NK) cells.

## 1. Introduction

There are three main subpopulations of T lymphocytes (T cells), all of which recognize antigens by T cell receptors (TCR). Cytotoxic CD8(+) T cells directly kill target cells primarily by secreting perforin and granzymes or via direct contact mediated by proteins belonging to Tumour Necrosis Factor Superfamily (TNFSF) members present on CD8(+) and their transmembrane receptors belonging to Tumour Necrosis Factor Receptor Superfamily (TNFRSF) present on target cells. Such an interaction results in triggering cytotoxic signals in target cells, although it must be remembered that certain TNF/TNFRSF members also have pro-inflammatory and pro-survival capabilities, depending on cellular context. The second population of T cells are helper CD4(+) T cells, whose primary function is to promote the activation of different arms of adaptive immune response by secreting different sets of cytokines, although they can also act as cytotoxic cells [1]. The third population are CD4(+)CD25(+)FoxP3(+) regulatory T cells that quench the immune response. All the mentioned populations of T cells can be further subdivided into different subpopulations, depending on their activity and sets of secreted cytokines. Chimeric antigen receptor T cells (CAR-T cells) are T cells genetically modified to express receptors recognizing antigens present on target cells to be eliminated. T cells are key cells of adaptive immunity. The process of T cells modification is random and different T cell populations could become functional CAR-T cells. Here, we would like to review T cells’ diversity in the context of CAR-T cells biology.

### 1.1. T Cell Receptor Signalling and Activation of T Cells

TCR receptors do not contain any signalling domains in their intracellular part. Their extracellular region contains three complementarity-determining regions (CDR) that directly bind antigens presented by major histocompatibility molecules (MHC) class I or II. Each TCR is non-covalently associated with three dimeric sets of CD3 molecules—CD3γε, CD3δε and CD3ζζ that contain immune tyrosine-based activation motifs (ITAM) and mediate signalling from TCR. Recognition of antigens presented in the context of MHC class I or II proteins is determined by the presence of CD8 or CD4 molecules, respectively, which also associate with the TCR complex. Binding of an antigen-MHC complex leads to phosphorylation of ITAM motifs on CD3ζ chains by Src or Lck kinases which subsequently bind ζ chain-associated protein kinase 70 (ZAP70) that phosphorylates ITAM motifs in other CD3 chains which now also can bind ZAP70. ZAP70 also phosphorylates other signal-transducing adaptor proteins, including PI3K and SLP76, which eventually results in the activation of transcription factors, such as NF-AT, NF-kB or AP-1 that activate expression of genes involved in promoting proliferation and effector functions of T cells [2,3].

Sole interaction between MHC-antigen complex and TCR is not enough to drive activation of naïve T cells. In fact, this interaction (often described as signal 1) results in anergy (state of inactivity) of T cells when delivered alone. For full activation, i.e., to achieve proliferation and acquire effector functions, T cells require at least one additional signal (signal 2) delivered by co-stimulatory molecules. Such molecules are usually divided into two main groups—the CD28-B7 family and TNF/TNFR superfamily [4,5,6].

### 1.2. Chimeric Antigen Receptor Signalling

CAR-T cells can be either CD4(+) or CD8(+). They contain transmembrane CAR receptor encoded by a genetic construct inserted into T cells usually by viral transduction, which in clinical practice is performed after isolation of T cells from the patient’s peripheral blood. CAR receptors are composed of a domain binding target antigen, which in most cases is a single-chain variable fragment (scFv) of an antibody, followed by hinge and transmembrane domains, optional domains encoding costimulatory molecule(s) and CD3ζ. CAR receptors are often classified into four generations depending on the structure of their intracellular domain—1st generation containing only CD3ζ, 2nd generation containing one costimulatory domain and CD3ζ, 3rd generation containing at least two costimulatory domains and CD3ζ, and 4th generation—also known as TRUCK (T-cells redirected for universal cytokine-mediated killing)—that apart from CD3ζ and co-stimulatory domains also contains transgenes for cytokines [7]. Binding between a CAR receptor and its ligand triggers the activation of co-stimulatory domains present in its structure and phosphorylation of ITAM motifs in the CD3ζ domain. CAR receptors do not require antigens to be presented on MHC molecules and can recognize them directly. Additionally, they are sufficient for activation of CAR-T cells, which do not need TCR receptors for their activity, although obviously there are functional differences between different types of CAR receptors. Interestingly, however, there are reports showing that even though TCR is dispensable for activation of CAR-T cells, its presence prolongs the persistence of CAR-T cells in vivo [8,9,10]. According to studies by Ramello et al. [11], 2nd generation CAR receptors spontaneously activate phosphorylation of ITAM motifs present in the CD3ζ domain and more intensively respond to antigen stimulation compared to 3rd generation CAR receptors as demonstrated by stronger phosphorylation of ZAP70 and Lck proteins [11]. This phenomenon depends neither on the type of hinge nor transmembrane domains, nor on the type of costimulatory domains. Instead, it appears to be dependent on the physical length of the hinge/transmembrane CAR domain. This certainly should be considered while designing CAR receptors, since constant, tonic activation of the CD3ζ domains in T cells can lead to their exhaustion [12].

## 2. CAR-T Cells Approved for Clinical Practice in the European Union

To date, the European Medicines Agency (EMA) approved for use in the European Union (EU) six types of 2nd generation CAR-T cells, which will be discussed in more details in subsequent sections of the article (Table 1): Kymriah (tisagenlecleucel, manufactured by Novartis Europharm Ltd.; agency product number EMEA/H/C/004090, approved on 22 August 2018) [13], Yescarta (axicabtagene ciloleucel; manufactured by Kite Pharma EU B.V.; agency product number: EMEA/H/C/004480, approved on 23 August 2018) [14], Tecartus (brexucabtagene autoleucel; manufactured by Kite Pharma EU B.V.; agency product number: EMEA/H/C/005102, approved on 14 December 2020) [15], Carvykti (ciltacabtagene autoleucel; manufactured by Janssen-Cilag International NV; agency product number: EMEA/H/C/005095, approved on 25 May 2022) [16], Abecma (idecabtagene vicleucel; manufactured by Bristol-Myers Squibb Pharma EEIG; agency product number: EMEA/H/C/004662, approved on 18 August 2021 [17], and Breyanzi (lisocabtagene maraleucel; manufactured by Bristol-Myers Squibb Pharma EEIG; agency product number: EMEA/H/C/004731, approved on 4 April 2022) [18].

### 2.1. Tisagenlecleucel

Tisagenlecleucel is approved in the EU for treatment of B-cell acute lymphoblastic leukaemia (B-ALL) in children and young adults up to 25 years of age and diffuse large B-cell lymphoma (DLBCL), and follicular lymphoma (FL) in adults with relapsing/refractory disease [13]. These CAR-T cells express the CAR receptor composed of the scFv fragment directed against CD19 antigen and derived from murine FCM63 monoclonal antibody, hinge and transmembrane domains derived from human CD8-a, followed by the 4-BB and CD3ζ domains [22,24]. Tisagenlecleucel T cells comprise of bulk, mixed CD4(+) and CD8(+) T cells which proportions are not specified [22]. A recent meta-analysis showed that tisagenlecleucel allows to achieve average overall response rate (ORR) of 53% for DLBCL, 86% for FL and 81% for B-ALL. Complete response rates (CRR) were about 40% for DLBCL, 73% for FL and 81% for B-ALL [19]. Depending on the laboratory technique used for detection of CAR-T cells, their persistence in patients reached 554 days (flow cytometry) or 693 days (quantitative PCR) post infusion [20].

### 2.2. Axicabtagene Ciloleucel

Axicabtagene ciloleucel is approved in the EU for treatment of adults with relapsing or refractory DLBCL, FL and primary mediastinal B-cell lymphoma (PMBL) [14]. CAR receptor in these cells is composed of FCM63 antibody-derived anti-CD19 scFv fragment, CD28-derived hinge and transmembrane domains followed by CD28 signalling and CD3ζ domains [22,24]. In contrast to other EMA-approved CAR-T cells, which are generated by lentiviral transduction, axicabtagene ciloleucel CAR-T cells are generated by transduction of T cells with retroviruses. Similarly to tisagenlecleucel, axicabtagene ciloleucel is composed of bulk T cells of unspecified CD4(+) to CD8(+) ratio [22]. Average ORR for DLBCL is 75%, for FL 81% [19], while for PMBL 80–73% depending of the type of patients’ pre-treatment [21,25]. CRR values were 52% for DLBCL, 64% for FL [19] and 72–54% for PMBL [21,25]. In a small study including three patients, Baras et al. [26] demonstrated the presence of axicabtagene ciloleucel CAR-T cells in patients’ blood up to 300 days post-infusion, as measured by both flow cytometry and quantitative PCR. Another study, conducted on a larger group of patients and including a much longer follow-up period, showed that the CAR-T cells remained present even up to 5 years post-infusion [25].

### 2.3. Brexucabtagene Autoleucel

Brexucabtagene autoleucel, approved by EMA for treatment of adults with mantle cell lymphoma (MCL) [15], has exactly the same CAR structure as axicabtagene ciloleucel. The difference is in the manufacturing process which in case brexucabtagene autoleucel involves the removal of CD19-expressing cells from the population of CAR-T cells [22]. In a three-year follow-up study ORR for patients with MCL was found to be 91%, while CRR was 68% [27].

### 2.4. Lisocabtagene Maraleucel

Lisocabtagene maraleucel is approved in the EU for treatment of adult patients with refractory and relapsing DLBCL, PMBCL, and FL grade 3B [18]. CAR receptor in these cells is composed of FCM63-derived anti-CD19 scFv fragment, CD28-derived hinge and transmembrane fragments followed by 4-1BB and CD3ζ domains [22]. This CAR-T cells preparation contains equal amounts of CD4(+) and CD8(+) T cells [22]. Average ORR for DLBCL is 72%, while CRR for DLBCL is 52% [19].

### 2.5. Ciltacabtagene Autoleucel

Ciltabtagene autoleucel is approved in the EU for treatment of adults with relapsing and refractory multiple myeloma. CAR receptor of these cells is directed against B cell maturation antigen (BCMA) and contains two camelid variable heavy chain antibody domains followed by 4-1BB and CD3ζ domains. The preparation is composed of bulk T cells [16]. CARTITUDE-1 study conducted on 97 patients showed very high values of ORR reaching 97%, and CRR of 78% [23].

### 2.6. Idecabtagene Vicleucel

Idecabtagene vicleucel is approved by EMA for treatment of refractory multiple myeloma. It is composed of bulk T cells which CAR receptor contains murine scFv fragment directed against BCMA and 4-1BB, and CD3ζ domains. The preparation may, however, also contain NK cells [17,23]. The clinical study KarMMa-1 conducted on 128 patients showed that idecabtagene vicleucel achieves 73% ORR, and 33% CRR [23].

## 3. Reasons behind Different Efficacy and Persistence of CAR-T Cells

Following infusion into a patient, CAR-T cells migrate into peripheral tissues and undergo dynamic phases of expansion and contraction. Usually, the number of CAR-T cells peaks after several to several dozen days after infusion and then begins to decrease [28], although CAR-T cells can be detectable in the patient’s blood even years post-infusion [25]. These changes, as well as the differentiation of CAR-T cells, affect the activity of CAR-T cells and do not depend solely on the structure of their CAR receptors, but also on the length and intensity of antigen stimulation, and the set of cytokines present in the microenvironment.

Another factor that affects the efficacy of CAR-T cells is their differentiation into effector or memory cells. Generally, effector T cells (T_EFF_) are characterized by quick response but low capability for self-renewal, short survival, and poor homing to tumour niches. On the other hand, central memory T cells (T_CM_) and stem-like memory T cells (T_SCM_) show lower cytotoxic capabilities than T_EFF_ cells but proliferate more rapidly and show sustained persistence in vivo [12].

Prolonged stimulation with an antigen often results in the exhaustion of T cells, which is a common cause of decreased CAR-T cells’ activity in vivo. The exhaustion of T cells is associated with the loss of their effector functions, decreased proliferation and the sustained expression of inhibitory molecules, such as PD-1/PD-L1, CTLA-4 or LAG-3 [12,29]. More prone to exhaustion are T_EFF_ rather than T_CM_ cells which, together with better in vivo persistence of the latter, is the cause of their higher anti-tumour potential [12].

Most EMA-approved CAR-T preparations, except for lisacabtagene maraleucel, contain a random mixture of CD4(+) and CD8(+) CAR-T cells, which complicates the analysis of treatment outcomes. Therefore, the composition of CAR-T cells’ preparation infused into patients possibly affects the efficiency of the therapy and only the products of precise cellular composition can provide uniform potency [30]. A case study by Rejeski et al. [31] showed that different subpopulations of CAR-T cells not only have different biological functions but also show distinct differentiation patterns. In the presented study, the phenotype of CD4(+) CAR-T cells shifted from central memory to effector type, while CD8(+) CAR-T cells’ phenotype changed from effector memory to terminal effector cells. Yet, at the same time, both subpopulations of CAR-T cells underwent equivalent clonal expansion [31].

One of the most obvious reasons explaining functional differences between CAR-T cells is the type of the co-stimulatory domain present in the CAR receptor. All CAR-T cells approved for use in the EU contain CAR receptors encoding either 4-1BB or CD28 co-stimulatory domain. Their roles and differences in the context of CAR-T cells’ biology are outlined below. However, the activity of CAR-T cells is also affected by the type of hinge and transmembrane domains present in the CAR construct. CAR-T cells with CD28-derived hinge and transmembrane domains form more stable immunological synapses and have lower threshold of activation then cells with CD8-derived domains [32].

### 3.1. 4-1BB

4-1BB (also known as CD137 or TNFRSF9) is the receptor for the co-stimulatory molecule 4-1BBL (also known as TNFSF9). In normal conditions, interaction between these molecules provides the 2nd signal required for activation of T cells. 4-1BB binds adaptor proteins TRAF1 and TRAF2 and downstream signalling from this receptor results in activation of AKT, p38 MAPK and ERK signalling pathways, which induce expression of several pro-survival proteins via NF-kB transcription factor, including anti-apoptotic Bcl-2 and Bcl-XL, as well as pro-inflammatory cytokines such as interferon gamma (IFN-γ) or interleukin 2 (IL-2), which also enhances proliferation of T cells [33,34,35]. Two mouse studies also showed that systemic administration of agonistic anti-4-1BB antibody results in the formation of cytotoxic T cells that are characterized with the elevated expression of granzymes, perforin and Fas ligand [36,37]. Interestingly, cytotoxic activity was not limited to CD8(+) T cells only but also was found in CD4(+) T cells [36]. These findings may be significant in the context of 4-1BB-encoding CAR constructs.

### 3.2. CD28

CD28 is a classical co-stimulatory receptor of T cells. It binds CD80 and CD86 ligands present on antigen-presenting cells. Following activation, signalling from CD28 leads to activation of several signal-transducing proteins, including PI3K, SLP76, ITK and VAV1, ultimately resulting in activation of pro-survival transcription factors, such as NF-AT and NF-kB. Interestingly, signalling pathways from TCR (more precisely, CD3ζ) and CD28 are integrated at the level of SLP76, VAV1 and ITK proteins, which become activated by both receptors [38]. Furthermore, ligation of CD28 in the absence of TCR signalling may result in triggering of inhibitory signals in T cells. Taking into account that CAR-T cells act independently of TCR, the presence of CD3ζ domain in CAR constructs may potentiate signalling from their CD28 domains and prevent the inhibition of activation.

### 3.3. Functional Differences between 4-1BB and CD28 CAR Domains

Many studies showed that co-stimulation provided 4-1BB and CD28 affects biology of T cells in different ways. Signalling from TCR and most TNFR superfamily members activates the canonical NF-kB pathway, which leads to rapid and transient activation of NF-kB and the nuclear translocation of canonical NF-kB heterodimer composed of RelA and p50 subunits. However, 4-1BB, but not CD28, signalling results in the activation of the non-canonical NF-kB pathway, which leads to slow and persistent activation of NF-kB, and nuclear translocation of the NF-kB RelB/p50 heterodimer. In the context of T cells’ biology, the canonical NF-kB pathway is associated with rapid cell expansion following their activation, while the non-canonical pathway is required for long-term maintenance of effector T cells and formation of memory T cells [39,40]. Indeed, 4-1BB-expressing CAR-T cells differentiate into CD45RO(+)CCR7(+) central memory cells and express high amounts of memory-associated transcription factors (incl. KLF6 and JUN), while CD28-expressing CAR-T cells differentiate into short-lived CD45RO(+)CCR7(-) effector cells [41,42]. Furthermore, 4-1BB-CAR-T cells show less exhausted phenotype following ex vivo expansion and express less inhibitory molecules, such as PD-1, CTLA-4 and LAG-3, compared to CD28-CAR-T cells [43,44]. 4-1BB-CAR-T cells also show better response to low-density antigens than CD28-CAR-T cells, which was demonstrated both in in vitro cytotoxic assays as well as in mouse tumour models [32]. This finding may be of significant importance, as one of the ways how tumour cells evade detection by T cells is internalization of antigens in the process known as trogocytosis. Even though both CD28- and 4-1BB-CAR-T cells are affected by this process, the latter are less prone to it due to their lower requirement for antigen density on target cells [45]. An in vitro study by Zhang et al. [46] showed that CD8(+) T cells expanded with 4-1BB co-stimulation show enhanced cytolytic activity compared to cells activated with CD28 co-stimulation. What is also important, is that there are reports showing that 4-1BB-CAR-T cells exhibit superior proliferation, in vivo persistence and tumor elimination compared to CD28-CAR-T cells [29,41].

## 4. Directions of CAR-T Cells’ Development

Even though CAR-T cell therapy proved to be successful in the treatment of haematological malignancies and could potentially be used for treatment of patients with solid tumours, it also has multiple limitations, such as toxicity, antigen escape, problems associated with tumour infiltration or immunosuppressive microenvironment. Cancer cells of haematological origin are easily identified and targeted by CAR-T cells based on the presence of cancer-specific antigens: CD19 or BCMA. However, lack of tumour-specific antigens in solid tumours leads to toxicity or tumour antigen escape. Even though certain markers of solid tumours, such as NKG2DL, GD2, EGFR, PD-1, CD171 are used in experimental therapies and studies on CAR-T cells directed against solid tumours, still new, more reliable markers of solid tumours are needed [47,48,49]. A recent study demonstrated the potential of new CAR-T cells targeting DCLK1 (CBT-511 cells). DCLK1 is a microtubule-associated kinase that regulates tumour growth and progression and is an antigen of tumour stem cells of colorectal cancer. The study showed that CBT-511 cells successfully eliminate colorectal cancer cells (HT29, HCT116) and induce secretion of IFN-γ [50]. Other studies proved the effectiveness of CAR-T cells targeting glypican-3 (GPC3), which is present in hepatocellular carcinoma (HCC) but not on normal liver tissue. Another obstacle in the process of development of effective CAR-T cells against solid tumours is the tumour microenvironment. For example, the hepatocellular carcinoma (HCC) microenvironment is devoid of interleukin (IL) 15 and 21, which are essential for the proper functioning of T-cells, therefore human GPC3-CAR T cells co-expressing IL-15 and IL-21 were very efficient in the treatment of HCC in preclinical models [51]. Targeting PD-L1-expressing cells might allow to overcome tumour escape from immune surveillance, therefore results of this study appear to be very promising, although further tests are still required. The treatment of solid tumours is associated with the problem of homing CAR-T cells into the tumour mass, which is not the case with haematological malignancies that are easier targets for CAR-T cells therapies. Nevertheless, other molecules, such as CD20, CD22, CD70, CD7 and CD5, could also be used to direct CAR-T cells to haematological targets [47,52].

### 4.1. Multi-Antigen-Targeted CAR-T Cells

A serious problem associated with CAR-T cell therapies is tumour antigen escape. This issue can be resolved by using two CAR receptors directed against different antigens and expressed in a single cell—a solution known as “dual CAR” [53]. A different strategy is to use tandem CAR receptors containing two distinct antigen-binding domains fused to one intracellular module [54,55]. Other solutions preventing antigen escape include the use of trivalent CAR-T cells and pooled CAR-T cells. The former contains three different CAR receptors (recognizing three distinct antigens) in a single T cell, while the latter is a combination of two or more CAR-T cells lines, each with different antigen specificity [49].

### 4.2. Inhibitory CAR-T Cells (iCAR-T Cells)

CAR-T therapy is associated with certain potentially serious adverse effects, including cytokine release syndrome, neurotoxicity, graft versus host disease (GvHD) and even cardiac arrest [22,56]. Therefore, apart from increasing the efficacy and target specificity of CAR-T cells, the better management and amelioration of adverse effects is one of the necessary directions for development of therapies based on CAR-T cells. Stenger et al. [10] showed that the knockout of TCR in CAR-T cells prevented their alloreactivity. Moreover, the activity of CAR-T cells can be regulated by inhibitory CAR-T cells (iCAR-T cells). This approach combines two chimeric receptors, one of which generates inhibitory signals for CAR-T cell responses. It contains an inhibitory receptor domain (PD1 or CTLA-4), which blocks the activity of T cell if the target cell is healthy (normal, unchanged antigen is expressed on the surface). The second receptor contains activating CD28 domain. Consequently, T lymphocytes are activated only when a tumour antigen is present on the cell surface—only the receptor with CD28 domain is involved in signalling. In this way, iCAR T cells can distinguish cancer cells from healthy cells and reversibly block the functionality of the transduced T cells in an antigen-selective manner [57].

### 4.3. SUPRA CAR-T Cells

To improve the controllability and universality of CARs, a SUPRA (Split, Universal, PRogrammAble) CAR system was designed. SUPRA CAR is composed of two main elements: universal receptor with leucine zipper adaptor (zipCAR) expressed on T cells that contains intracellular signalling domains and a transmembrane domain fused to an extracellular leucine zipper; the other element is a protein construct, a separate antigen-binding scFv fragment bound to another leucine zipper (zipFv). Consequently, the CAR receptor gains activity only after binding to the zipFv. This system allows for fine tuning and calibration of CAR-T cells’ antigen specificity and activity by regulating extracellular concentration of zipFv proteins of different antigen specificity or addition of zipFv proteins that dimerize with other, thereby down-regulating the activity of CAR-T cells. The SUPRA CAR system showed a reliable anti-tumour effect comparable to that of the classical CAR-T cells in mouse xenograft models of breast cancer and leukaemia [58].

### 4.4. Universal CAR-T (UCAR-T) Cells

Another serious disadvantage of classic CAR-T therapy is that CAR-T cells have to be every time and for every patient developed from the patient‘s own cells. This approach drastically lengthens time and rises costs of CAR-T therapy. To overcome this obstacle, the construct of tumour antigen-specific T cells from allogeneic healthy donors (universal CAR T cells) which do not trigger harmful responses—graft versus host disease (GvHD) and host versus graft disease (HvGD)—was proposed [59]. Universal CAR-T cells provide fast and profitable CAR production [60].

### 4.5. CAR-NK Cells

Another approach to overcome GvHD and HvGD is the expression of CARs in natural killer (NK) cells. These lymphocytes can target tumour cells without prior antigen priming and human leukocyte antigen (HLA) matching. They function as allogeneic effectors and do not need to be collected from a specific HLA-matched donor, and therefore are suitable for the development of an “off the shelf” therapeutic product [61]. Their cytotoxic activity is regulated by the integration of signals delivered by activating and inhibitory receptors expressed on their cell surface [62]. Thus, therapy with CAR-NK cells poses a minimal risk of GvHD or HvGD development. CAR-NK cells also rarely cause cytokine release syndrome (CRS) and immune cell-associated neurotoxicity syndrome (ICANS) [63]. Functional natural killer cells can be obtained from different sources to generate CAR-NK cells, including peripheral blood (PB) [64], NK cell lines [65], umbilical cord blood (UCB) [66], placental blood [67], human embryonic stem cells (hESCs) [68] and induced pluripotent stem cells (iPSCs) [69,70].

Similar to CAR-T cells, CAR constructs in NK cells are composed of an extracellular antigen recognition region, a hinge region, a transmembrane region and one or more intracellular signalling domains. Initially in CAR-NK studies were used constructs designed for CAR-T cells with intracellular domain composed of CD3ζ and costimulatory domains containing CD28, CD137 (4-1BB), ICOS, CD27, OX40 or CD40. Then, novel constructs designed more specifically for CAR-NK cells were introduced with transmembrane NKG2D and DAP10, DAP12 or 2B4 signalling domains [71]. The natural cytotoxicity of NK cells allows CAR-NK cells to mediate their cytotoxic activity in both CAR-dependent, based on CAR-mediated direct cytotoxicity, and CAR-independent manner, involving NK cell natural killing mechanism [72].

Multiple preclinical and clinical trials have been carried out to evaluate the efficacy of CAR-NK cells for B cell malignancies with target antigens frequently used also in CAR-T therapy for B-cell lymphoma and leukaemia, i.e., CD19, CD20, CD22 [63,71,73]. Several clinical trials also concerned T cell malignancies with target antigen CD7 [74]. Then, certain in vitro and in vivo studies were conducted on target antigens CD5 for T cell malignancies and CS1 and CD138 for multiple myeloma [75]. CAR-NK cells were preclinically tested in multiple solid tumours including breast cancer, ovarian cancer, pancreatic cancer, colorectal cancer, hepatocellular carcinoma, renal cell carcinoma, glioblastoma, osteosarcoma and melanoma [73,76]. However, clinical trials conducted on this group of cancers were very limited and regarded pancreatic cancer, epithelial ovarian cancer, prostate cancer, glioblastoma and non-small cell lung cancer [63,73].

### 4.6. Genomic Targeting of CAR Constructs

According to standard protocols, lymphocytes are genetically modified to express chimeric antigen receptors with viral vectors, especially lentiviral vectors are optimal to obtain efficient transfection and stable gene expression. However, incidental genome integration may lead to mutation and/or disruption of other functional genes. So that, gene-editing methods are being explored to ensure efficient transfection, reduced GvHD and enhanced expression of CARs. Techniques such as zinc-finger nucleases (ZFN), transcription activator-like nucleases (TALENs) and CRISPR/Cas9 have been used for preparation of CAR-T cells [60]. Undoubtedly, this opens novel perspectives to develop next generation of CAR-T cells targeting specific genomic loci.

## 5. CAR-T Therapy in Poland

### 5.1. Current State of CAR-T Therapy in Poland

Polish studies on CAR-T cells and current state of CAR-T therapy are not as advanced as in most European Union member states. In Poland, CAR-T therapy was used for the first time in the “Cape Hope” University Clinic in Wrocław, in 2020 [77]. The patient was an 11 years old boy with acute lymphoblastic leukaemia who had previously received conventional therapies including two allogenic bone marrow transplants that had not ended in a success, therefore CAR-T cells were a last chance therapy for him. One year after treatment with CAR-T cells, the patient did not show any signs of minimal residual disease. All CAR-T cell therapies approved in the EU are also approved in Poland. However, only tisagenlecleucel and axicabtagene ciloleucel are reimbursed in all approved indications, except for follicular lymphoma. Specifically, starting from 1 September 2021, CAR-T (tisagenlecleucel, Kymriah) therapy is reimbursed by the Polish National Health Fund for relapsing/refractory acute lymphoblastic leukaemia in children and adults up to 25 years of age. Furthermore, from May 2022, CAR-T treatment costs can also be covered for patients with large B-cell lymphoma treated with Kymriah or Yescarta (axicabtagene ciloleucel) and large B-cell lymphoma treated with Yescarta in a case of failure of at least two lines of treatment. Currently, there are six medical centres in Poland that are authorized to use CAR-T cell therapy—(1) Clinic of Haematology and Bone Marrow Transplantation, Medical University in Poznań, (2) M. Skłodowska-Curie National Research Institute of Oncology branch in Gliwice, (3) Clinic of Bone Marrow Transplantation, Oncology and Paediatric Haematology “Cape Hope”, University Clinical Hospital in Wrocław, (4) Clinic of Haematology, Transplantology and Internal Diseases, University Clinical Centre of Warsaw Medical University in Warsaw, (5) Clinic of Haematology and Transplantology, Medical University of Gdańsk in Gdańsk and (6) Voivodeship Multi-Specialist Centre for Oncology and Transplantology in Łódź. It is estimated that 15–20 paediatric patients will be qualified for CAR-T cells therapy in Poland each year, which equals to an approximate cost of more than 4–5 million euro (EUR).

### 5.2. Current Studies and Future of CAR-T Cells in Poland

The high costs of commercial CAR-T cells and the limitations of the reimbursement program hamper the development of CAR-T cell therapy in Poland. As Talha K. Burki [78] pointed out in a recent commentary, the key factor for low- and middle-income countries might be onsite production of CAR-T cells for local use. One of the solutions is the development of “academic CAR-T cells” as it was successfully done in Spain [79,80], Turkey [81,82] or Switzerland [83]. Such cells are developed and produced in academic units and tailored for local (national) requirements. This approach allows for the decentralized production of CAR-T cells, which can be prepared in the same unit where patients are hospitalized, allowing to achieve faster turnover time compared to currently available commercial CAR-T cells. What is also important, is that “academic CAR-T cells” have potential for commercialization, which is especially important for countries like Poland where no commercial companies are currently involved in large-scale CAR-T cells’ development. Cooperation between academic units and commercial companies can improve the country’s economy and increase competition in the market of anti-tumour therapies, eventually leading to a drop in prices of CAR-T therapies. Finally, such an approach significantly boosts research on new solutions in CAR-T therapy, aiming to overcome its current limitations and to expand its application to additional types of cancer. Thus, the progress in research on locally developed and produced “academic CAR-T cells” and the extension of their application to new tumour subtypes, including solid tumours, is necessary for the dissemination and better availability of cell-based therapy, not only in Poland, but also throughout Europe.

At the present time, studies on CAR-T cells in Polish academic units are at the beginning of their way, and Poland has a very high potential for both the development and commercialization of CAR-T solutions. Certain universities are currently conducting studies aiming to identify new targets for CAR-T cells and to modify scFv fragments of CAR receptors to better recognize cancer antigen [84]. There are also ongoing basic studies on the biology of CAR-T cells, although no data have been published to date, except for a Polish-Norwegian research team that recently described CAR-T cells targeting inhibitory ligand PD-L1 present on cancer cells and other cells in the tumor microenvironment. These cells were shown to kill PD-L1(+) cells both in vitro and in a mouse model [85].

The Polish government recently issued a “Governmental plan for development of biomedical sector in years 2022–2031” with the intention to achieve a leading position in biomedical sector Central-Eastern Europe [86]. The plan is being implemented by the Polish Medical Research Agency, which in 2020 has awarded a grant worth 25 million EUR to a consortium “Polish Chimeric Antigen Receptor T-Cell Network” composed of eight research and clinical facilities, with the leading role of Medical University of Warsaw, Poland. The consortium intends to start a clinical study “MERMAID1” involving Polish-made CAR-T cells used for the treatment of 180 patients with haematological malignancies in years 2024–2025. The estimated cost of a single CAR-T preparation is ca. 32′000 EUR which is around nine times lower than cost of currently available commercial CAR-T cells [87]. Additionally, Medical University in Wrocław, Poland, is currently working to develop a technological platform for production of CAR-T cells in Poland [88]. Thus, the development of academic CAR-Ts seems to be an attractive option in Poland to provide inexpensive but efficient cell-based therapy.

## 6. Conclusions

Even though several types of CAR-T cells are already used in clinical practice, often with very good outcomes, they still need further improvements. Additionally, the biology of CAR-T cells certainly still requires better understanding. Even slight variations between formulations of CAR-T cells preparations or the construction of CAR receptors can result in changes in biology of these cells, and consequently in the effects of treatment. For example, the only difference between axicabtagene ciloleucel and brexucabtagene autoleucel lies in the manufacturing process and the removal of CD19(+) cells from the latter type of CAR-T cells. These two types of CAR-T cells are, however, approved for the treatment of different types of cancer. Additionally, seemingly insignificant changes in the CAR structure, such as different types of hinge and transmembrane regions, lead to changes in the activation of CAR-T cells [32]. Therefore, medicine based on CAR-T cells strongly needs more in-depth studies on activation, differentiation, expansion, and the exhaustion of these cells, which would allow to invent more specific and more efficient CAR-T cells. We believe that the future of CAR-T cell therapy in Poland and countries of similar economic status lies in efficient solutions developed in local academic units, which can be implemented into clinical practice and commercialized to increase competition in the market, and eventually to decrease the price of CAR-T therapy.

## Figures and Tables

**Table 1 biomedicines-10-02912-t001:** CAR-T cells approved for clinical practice in the European Union.

Medication	Treated Condition	CAR Construct	References
Tisagenlecleucel (KYMRIAH, Novartis Pharmaceuticals Corporation (Basel, Switzerland))	B-cell acute lymphoblastic leukaemia (B-ALL) in children and young adults up to 25 years of age diffuse large B-cell lymphoma (DLBCL) in adults with relapsing/refractory cancers follicular lymphoma (FL) in adults with relapsing/refractory cancers	anti-CD19 extracellular domain 4-1BB (CD137) costimulatory domain CD3ζ intracellular signaling domain	[13,19,20]
Axicabtagene ciloleucel (YESCARTA, Kite Pharma Inc. (Los Angeles, CA, USA))	adults with relapsing or refractory DLBCL, FL and primary mediastinal B-cell lymphoma (PMBL)	anti-CD19 extracellular domain CD28 costimulatory domain CD3ζ intracellular signaling domain	[14,19,21]
Brexucabtagene autoleucel (TECARTUS, Kite Pharma, Inc.)	adults with mantle cell lymphoma (MCL)	almost the same CAR structure as axicabtagene ciloleucel	[15,22]
Lisocabtagene maraleucel (BREYANZI, Juno Therapeutics, Inc. (Seattle, WA, USA), a Bristol-Myers Squibb Company)	adult patients with refractory and relapsing DLBCL, PMBCL, and FL grade 3B	anti-CD19 scFv fragment CD28 transmembrane domain 4-1BB (CD137) costimulatory domain CD3ζ intracellular signaling domain	[18,19]
Ciltacabtagene autoleucel (CARVYKTI, Janssen Biotech, Inc. (Horsham, PA, USA))	adults with relapsing and refractory multiple myeloma	two BCMA-targeting domains 4-1BB (CD137) costimulatory domain CD3ζ intracellular signaling domain	[16,23]
Idecabtagene vicleucel (ABECMA, Celgene Corporation (Summit, NJ, USA), a Bristol-Myers Squibb Company)	refractory multiple myeloma	extracellular BCMA-targeting domain CD8α hinge 4-1BB (CD137) costimulatory domain CD3ζ intracellular signaling domain	[17,23]

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
