# Peer review of "Current and Future Perspectives for Chimeric Antigen Receptor T Cells Development in Poland"

_biomedicines, 2022, doi:10.3390/biomedicines10112912_

Round 1

Reviewer 1 Report

The manuscript “Current and future perspectives for chimeric antigen receptor T cells development in Poland” is generally well written and presents an overview of the CAR-T cell literature, starting specifically with T cell signaling, proceeding to describe CAR-T cell structure and different generations of CAR T cells, and then very briefly the difference in their signaling to that of endogenous T cells. Authors then discuss the different CAR-T cell products currently approved for clinical use in the European Union, including a brief description of each product, their indications, and their clinical efficacy. They then talk about the status of CAR-T cell therapy in Poland and some of the challenges to widespread use of CAR T cells for treatment of cancer patients in Poland. They further propose some solutions to current challenges, which if implemented successfully could be an answer to providing high quality cancer treatment strategies for not just Poland but other countries facing similar challenges. Following this, the authors go back to discussing the efficacy of CAR-T cells, the reasons for the difference in efficacy of CAR-T cells and end the manuscript with future directions in CAR-T cell development.  While all sections of the manuscript are appropriate and contain sufficient information, the overall structure of the manuscript could be improved.  Sections 2 & 3 are seemingly out of place, given that sections 1, 4, & 5 provide a more general review of CAR T cell therapy and are broken up by specific information about CAR T cell use in the EU and Poland. Reorganization of the manuscript could improve its flow and readability. Additionally, additional elaboration on overcoming challenges to CAR T cell therapy in Poland would improve the manuscript.

Author Response

Dear Reviewer,

Thank you for the careful review of our manuscript. According to your suggestions, we have reorganized sections of the manuscript to make it better structured. We have re-written and expanded the section about the current state of therapy and studies on CAR-T cells in Poland and improved the discussion about the future of CAR-T cells development in Poland (Section 5). In addition, we have modified the section about SUPRA CAR-T cells (Section 4.3) to make it more clear and corrected some minor linguistic errors. Our corrections are presented in a red font.

On behalf of all authors, kind regards,

Tomasz Åšlebioda

Reviewer 2 Report

The authors discussed the development and clinical status of CAR-T and also mention about current status of CAR-T in Poland.

1. Though the title is "Current and Future Perspectives for Chimeric Antigen Receptor 2 T Cells Development in Poland", there were less information on the status of Poland. The authors should focus more on the developing status of CAR-T in Poland, the strategy, the difference and difficulties of CART in Poland, as the title mention.

2. The authors should have more titles and/or figures to highlight the issues they discussed.   

Author Response

Dear Reviewer,

Thank you for the review of our manuscript. According to your suggestions, we have re-written and expanded the section about the current state of therapy and studies on CAR-T cells in Poland and improved the discussion about the future of CAR-T cells development in Poland (Section 5). We also divided this part into two subsections to make it clearer. Unfortunately, our capabilities do not allow us to add illustrations of proper quality in such a short time. In addition, we have reorganized sections of the manuscript to make it better structured and modified the section about SUPRA CAR-T cells (Section 4.3) and corrected some minor linguistic errors. Our corrections are presented in a red font.

On behalf of all authors, kind regards,

Tomasz Åšlebioda

Reviewer 3 Report

This is a comprehensive and well-written review on CAR-T cells.

I have two suggestions for the authors:

1. In line 182 they need to explain what they "academic CAR-T cells" as it's not a commonly used term and one of the solutions they propose

2. The conclusion needs to have a line or two referring to Poland as this is the title of the manuscript

Author Response

Dear Reviewer,

Thank you for the review of our manuscript. According to your suggestions, we have explained the term “academic CAR-T cells” and expanded the section about the current state of therapy and studies on CAR-T cells in Poland and improved the discussion about the future of CAR-T cells development in Poland (Section 5). We also added the appropriate summary in the conclusion. In addition, we have reorganized sections of the manuscript to make it better structured and modified the section about SUPRA CAR-T cells (Section 4.3) and corrected some minor linguistic errors. Our corrections are presented in a red font.

On behalf of all authors, kind regards,

Tomasz Åšlebioda

Round 2

Reviewer 2 Report

The authors responded and modified the manuscript well accordingly to reviewers' comments.